# Evaluating the Impact of Regulatory Guidelines on Market Adoption and Implementation of Telehealth for COPD Patients: A Systematic Literature Review

**DOI:** 10.3390/healthcare13222858

**Published:** 2025-11-11

**Authors:** Noha Saeed Alghamdi, Nora Ann Colton, Paul Taylor

**Affiliations:** Global Business School for Health, University College London, London WC1E 6BT, UK; n.colton@ucl.ac.uk (N.A.C.); p.taylor@ucl.ac.uk (P.T.)

**Keywords:** COPD, telehealth, regulatory guidelines, digital health, systematic review, implementation barriers, healthcare policy

## Abstract

**Purpose:** Telehealth (TH) offers promising solutions for enhancing the management of chronic obstructive pulmonary disease (COPD), particularly in resource-limited or remote settings. However, regulatory uncertainty remains a significant barrier to adopting and integrating TH technologies into routine care. This systematic review aims to evaluate the role of regulatory guidelines in implementing and adopting TH solutions for COPD care and to identify key barriers and facilitators shaping these regulatory efforts. **Methods:** Following PRISMA guidelines, a comprehensive search of five databases up to 18 October 2025 (PubMed, Web of Science, Scopus, CINAHL, and JSTOR) and grey literature was conducted. Studies and governmental reports were included if they examined regulatory frameworks, stakeholder perspectives, or implementation challenges related to TH in COPD care. Study quality was assessed using the Critical Appraisal Skills Programme (CASP) tool. Narrative and data synthesis were employed. **Results:** From 343 identified records, 33 sources (18 peer-reviewed studies and 15 governmental/organizational reports) met the inclusion criteria. Findings revealed wide disparities in the existence, specificity, and enforcement of TH regulatory guidelines across countries. Developed nations often had more structured yet nonspecific frameworks, while emerging health systems, such as Saudi Arabia, exhibited fragmented but evolving regulatory landscapes. Common barriers included unclear stakeholder roles, inadequate funding, technological limitations, and resistance to organizational change. **Conclusions:** Clear, inclusive, and context-sensitive regulatory guidelines are essential to support the successful integration of TH in COPD care. Enhanced regulatory clarity can improve patient trust, engagement, and adherence by addressing safety, accountability, and accessibility concerns. Future research should focus on stakeholder-informed policies that reflect the practical realities of healthcare delivery in both developed and emerging systems.

## 1. Introduction

Chronic Obstructive Pulmonary Disease (COPD) is a progressive respiratory illness characterized by airflow limitation, commonly caused by prolonged exposure to harmful particles or gases, particularly tobacco smoke and air pollution [1]. Globally, COPD represents a significant public health burden. Managing COPD effectively requires an integrated, patient-centered approach that addresses not only pharmacologic treatment but also behavioral support, pulmonary rehabilitation, and regular monitoring. Telehealth (TH) is a collective term encompassing a range of digital technologies used to deliver and monitor healthcare remotely, and it has emerged as a promising tool in this context [2,3]. By leveraging information and communication technologies (ICTs), TH enables continuous, personalized care delivery to patients with chronic illnesses such as COPD, especially those living in underserved or remote areas [4]. Since not all TH solutions available in the market qualify as medical devices, TH overlaps with related concepts such as telemedicine, remote patient monitoring, tele-respiratory care, tele-nursing, and tele-pulmonary rehabilitation. This interchangeable usage reflects the expanding landscape of digital healthcare delivery [5,6,7].

Despite the rapid development of TH globally, the adoption of TH into routine COPD care remains inconsistent. Multiple factors contribute to this limited uptake, including unclear regulatory pathways, lack of organizational readiness, stakeholder resistance, and an absence of standardized implementation frameworks [8,9,10]. While the COVID-19 pandemic catalyzed an unprecedented expansion of TH services, the long-term sustainability of these services remains uncertain [3]. In Saudi Arabia, for instance, the Ministry of Health (MoH) launched several digital initiatives during the pandemic to strengthen healthcare delivery [11]. However, researchers have noted that these initiatives often lacked robust regulatory frameworks, suffered from fragmented implementation, and failed to include input from key stakeholders such as frontline clinicians and patients [12]. The absence of user engagement, particularly from those directly involved in care delivery, may hinder the long-term use of TH, as successful TH solutions must address end-user needs. Another barrier to the effective adoption of TH solutions in COPD care is the absence of clear regulatory guidelines that govern the initiation, operation, and management of TH systems within healthcare settings. Without such frameworks, healthcare providers may face uncertainty, inconsistent practices, and legal or ethical concerns, all of which can undermine confidence in TH and limit its integration into routine clinical care.

Central to the successful integration of TH into COPD care is the establishment of clear regulatory guidelines. Regulations in the healthcare context may take various forms, including guidelines, frameworks, policies, and legislation. These structures are essential for ensuring the safety, quality, and efficacy of TH interventions, especially TH interventions involving medical devices or the remote delivery of care [13]. Regulatory guidelines also provide clinicians with a standardized reference for adopting TH tools, thereby promoting consistency and reducing variations in care.

However, the literature highlights significant gaps in knowledge, practical, and culturally appropriate regulatory frameworks tailored to TH use in COPD care, particularly in developing and emerging health systems [13,14,15,16]. Existing regulations and guidelines are generic and lack disease-specific guidance. Empirical research on real-world regulatory guidelines implementation is limited, and the influence of infrastructure and organizational context remains unclear. Moreover, no consensus exists on the most suitable theoretical framework for guiding TH implementation in COPD care.

Therefore, this systematic literature review seeks to address these gaps by synthesizing the available literature on the regulatory aspects of TH implementation for COPD care, with a specific focus on developing and emerging healthcare systems. The literature review aims to: (1) explore the role of regulatory guidelines in shaping implementation and assess the extent to which current frameworks are responsive to contextual, organizational, and patient-level needs; (2) identify the existing barriers and facilitators associated with regulatory guidelines governing the use of TH in COPD care.

## 2. Materials and Methods

### 2.1. Search Strategy and Selection Process

This method was developed as part of doctoral research, which adhered to the Preferred Reporting Items for Systematic reviews and Meta-Analyses (PRISMA) guidelines [17]. The study protocol was registered with PROSPERO under registration number CRD420251178887. We conducted an electronic search on the five databases, including sources for grey literature: Web of Science, PubMed, CINAHL, Scopus, and JSTOR. Sources of gray literature are also included in the search procedures to retrieve any relevant sources, such as governmental reports up to 18 October 2025. The search was conducted with supervision and guidance by an expert librarian at the UCL library (Name: Heather Chesters). Search strategies from all databases were provided in Table 1 and Appendix A.

The screening process was conducted in accordance with the PRISMA 2020 guidelines [17]. Initially, the authors, NA and PT, independently screened titles and abstracts to exclude studies that did not meet the inclusion criteria. In the second stage, the remaining articles underwent a full-text review to determine final eligibility. Any discrepancies between reviewers (NA and PT) during either stage were resolved through discussion or consultation with a third reviewer (NC).

### 2.2. Eligibility Criteria

#### Inclusion Criteria and Exclusion Criteria

The criteria for including articles in this systematic review focused on studies that examined regulatory guidelines and the factors influencing the implementation of TH in COPD care. However, the review identified a scarcity of such studies. To the best of our knowledge, no scientific research has specifically addressed the implementation of regulatory guidelines for TH in patients with COPD. Therefore, the scope of our electronic search was expanded to encompass governmental and non-governmental reports that discuss regulatory guidelines for TH in COPD care. The inclusion criteria were presented in Table 2.

### 2.3. Data Extraction and Quality of the Studies

We developed a standardized Microsoft Excel spreadsheet to systematically collect data from the included articles. The data extraction form was designed to capture information from qualitative studies and policy or guideline documents. It focused on elements related to TH regulations and frameworks in the context of COPD care. Extracted data included study characteristics, descriptions of TH applications, regulatory guideline content, implementation models, and identified institutional, user-level, and industry-level barriers. To ensure the methodological quality of the included studies, we applied the Critical Appraisal Skills Programme (CASP) checklist [18,19]. The CASP tool is widely recognized in evidence-based research for its structured approach to evaluating the validity, results, and relevance of qualitative and quantitative studies [18,19]. This tool was chosen due to its comprehensive nature, covering key domains such as methodological rigor, validity of findings, and relevance to the research question. In addition, grey literature was evaluated using Authority, Accuracy, Coverage, Objectivity, Date, and Significance (AACODS) checklist [20]. However, it is important to note that the quality appraisal was conducted independently of the study selection process. No studies were included or excluded based on their quality scores; rather, the appraisal was used to inform the interpretation and synthesis of the evidence [18].

### 2.4. Narrative and Data Synthesis

For the data analysis, we employed a narrative synthesis approach to systematically explore, organize, and interpret the extracted findings. Also, a quantitative frequency tables were applied to report barriers and facilitators identified across included studies. Then, a comparative thematic synthesis was conducted to explore differences between high-income and low-/middle-income health systems, allowing contextual interpretation of regulatory, infrastructural, and organizational variations influencing TH implementation for COPD care. A meta-analysis was not conducted, as the included studies were qualitative in nature and exhibited heterogeneity in design and outcome reporting. The included studies were summarized and synthesized under subheadings into two main groups:Published Articles, which were further summarized under categories into:
Descriptions of TH UseRegulatory Guidelines for TH Adoption in COPD CareIdentified Needs for Regulatory GuidelinesIntegration of TH into Health SystemsResponsibilities Specified in GuidelinesImplementation FrameworksReported Barriers and FacilitatorsGovernmental and Organizational Reports, which were summarized and critically appraised based on the region or organizational source.

## 3. Results

### 3.1. Screening and Studies Selection

The electronic search of academic databases and grey literature sources yielded a total of 343 records, including research articles and government reports. After removing 87 duplicate entries, 226 records remained for title and abstract screening. Based on the predefined inclusion and exclusion criteria, 46 records and 20 reports were deemed eligible for full-text review, comprising 18 peer-reviewed research articles and 15 government or organizational reports. All references were managed using EndNote software (version 20) to ensure accurate citation and organization. The selection process was systematically documented and illustrated using a PRISMA flow diagram (Figure 1).

### 3.2. Study Characteristics

The systematic literature review included 33 records, ranging from the oldest article published in 2008 to the most recent one in 2022. Regarding the governmental reports, the oldest report was in 2015, and the latest was in 2024. Among all included articles, the countries where regulatory guidelines were discussed are the United Kingdom (n = 7), Canada (n = 4), the Netherlands (n = 3), China (n = 2), Italy (n = 2), Germany (n = 2), Scotland (n = 1), Japan (n = 1), Ireland (n = 1), Greece (n = 1), Nepal (n = 1), Spain (n = 1), Denmark (n = 1), and Norway (n = 1) (Figure 2). All included studies were qualitative studies with semi-structured interviews with different stakeholders, such as managers, administrators, informants, healthcare providers, patients, or people from industries summary of the included studies provided in Table 3. The stakeholders who directly impact implementation or market adoption of TH solutions were recruited in 13 qualitative studies out of 18.

### 3.3. Descriptions of TH Use Studies

Among all included studies, terminologies to describe the use of technology or new biomedical devices in COPD care were interchangeable between TH [21,23,24,25,26,27,30,32,35,37,38], digital health [31,33,34,36], and telemedicine [22,28,29]. Despite the terminology variation, all reviewed TH solutions, regardless of the label used, shared a common goal which is enable remote monitoring and management of patients with COPD. This included functionalities such as tracking symptoms, transmitting data such as vital signs, and supporting remote clinical decision-making. The interchangeable use of terms reflects the lack of standardized definitions in the digital health landscape, especially related to regulatory guidelines.

### 3.4. Regulatory Guidelines for TH Adoption in COPD Care

Studies have demonstrated a variation in the existence of regulatory guidelines for the adoption of TH in the management of COPD care within local practices across European countries, United Kingdom (including Wales and Scotland) [21,22,24,26,28,30,32,37], Ireland [31], Germany [28,37], Spain [28], Greece [32], Denmark [32], Norway [32], Netherlands [23,25], and Italy [29,37]. In Asian countries, China and Nepal have a regulatory guideline for adopting TH in COPD care [35,36,38].

In the included studies, regulatory guidelines commonly addressed key components such as data protection, remote patient monitoring protocols, and clinical governance. Notably, developed healthcare systems, particularly in Europe, demonstrated more advanced integration of TH into COPD care, supported by well-established regulatory guidelines and cross-regional research projects [28].

In contrast, emerging healthcare systems in Asia have shown initial progress in formulating regulations to support the adoption of TH for COPD management [35,36,38]. Although these regulatory efforts are relatively recent, they indicate a growing governmental recognition of TH as a strategic response to tackle the increasing burden of chronic diseases.

### 3.5. Identified Needs for Regulatory Guidelines

The included studies demonstrated several factors that can affect regulatory guidelines for TH applications in COPD care. The included studies have a growing sense of the need to establish and adopt TH solutions in COPD care. For example, between 2008 and 2012, there was an increasing demand for community-based care for chronic conditions such as COPD [21,22,23]. Evidence from this period demonstrated a pressing need to create regulatory guidelines for TH usage in primary care settings, particularly with COPD patients [21,22,23].

Since 2017, there has been a noticeable shift in focus toward ensuring the sustainability of TH interventions in the management of COPD [29,38]. Increasingly, research has recognized the critical role of regulatory frameworks, policies, and legislation in supporting the long-term viability and integration of TH solutions into healthcare systems. To ensure the sustainable implementation and adoption of TH services, developers and policymakers should consider local factors and financial constraints specific to the healthcare system [21,22,23].

Additionally, the needs for regulatory guidelines and frameworks in the current healthcare systems vary from one location to another [21,22,27,32]. In developed health systems where COPD care guidelines were a priority, the provision of regulating new TH services is much faster than in other health systems. Summary of frameworks was summarized and synthesized in Table 4.

### 3.6. Integration of TH into Health Systems

Changing the health system by integrating TH into routine care practices could be a factor impacting the regulatory guidelines [23]. Also, policies and regulations of using TH must not rely only on the feasibility and effectiveness of TH solutions, including stakeholders’ needs and demands, which are important to design the guidelines [25]. Champion stakeholders (Leaders, policymakers, or payers) are required to initiate and lead the change in the health system, either in a particular project related to TH solutions or by changing the policy and regulations related to the use of TH [24,26,29,32].

#### 3.6.1. Responsibilities Specified in Guidelines

Most of the included studies demonstrated that the current guidelines for TH are general and not specific to frontline staff, such as nurses, respiratory therapists, and physicians [22,23,24,26,27,29,31,32,33,35,36,38] or even payers such as industrial or insurance companies [25,30]. To ensure the sustainability of regulatory guidelines for adopting TH in COPD care, specific tasks and responsibilities must be clarified for targeted frontline stakeholders, such as nurses, respiratory therapists, and physicians [25,29,30,34,35,37]. This will enhance cooperation within hospitals and between hospitals and primary care structures, as well as keep each level informed about their tasks and responsibilities [25,29,30,34,35,37]. This includes the admin responsibilities and industrial responsibilities within the regulatory guidelines [25,26].

#### 3.6.2. Reported Barriers and Facilitators

The regulatory guideline of TH faces numerous barriers that hinder its effective implementation and sustainability [29]. One of the primary challenges is resource constraints, including limited funding, workforce shortages, and infrastructure issues. These constraints make it difficult to establish and maintain guidelines for TH services. Additionally, organizational barriers contribute to fragmented communication and a lack of coordination between different policymakers, further complicating the integration of TH into existing health systems. Policy and regulation implementation also poses a significant challenge, as translating policies and regulations directly into practical service can be complex and time-consuming. Resistance to change among staff is another major barrier, with many healthcare professionals hesitant to adopt new working methods [29].

TH projects highlight additional obstacles. It often lacks clear clinical objectives, making it difficult to measure TH effectiveness. TH devices often lack practicality for sustained patient use, and data collection and analysis procedures are frequently manual rather than automated. Moreover, the implications of new interprofessional relationships and the medico-legal responsibilities associated with TH have not received sufficient attention, leading to confusion and reluctance among healthcare providers to adopt and accept TH [22].

Incomplete data capture and the absence of team dynamics also hinder the success of TH initiatives [26]. The creation and deployment of TH systems incur significant costs in terms of time and resources, necessitating project leaders (often referred to as Champions) to carefully direct these initiatives [32]. TH services often have restricted referral pathways, leading to consideration primarily for patients with significant medical needs. However, challenges remain in accurately assessing patient suitability and predicting the effects of TH on health outcomes. Staff reservations about using new technologies, concerns about increased workload, and the impact on nursing roles further complicate the adoption of TH [26].

Additionally, the regulatory guidelines frequently lack clarity regarding the duration of TH and the discontinuation process in cases where patients become dependent on remote monitoring [26,28]. Healthcare practitioners do not always prioritize TH. Finally, the absence of a shared vision and strategic ownership for investing in TH technologies further slows down the adoption of TH [26,28]. A list of 16 barriers and 16 facilitators has been identified from the included studies described in Figure 3 and Figure 4. A summary of the frequency and percentage distribution of identified barriers and facilitators is presented in Appendix A.

### 3.7. Quality of Included Studies

The methodological quality of the included studies was assessed using the CASP checklist. Overall appraisal scores ranged from 60% to 100%, indicating moderate to high methodological rigor. Two studies scored 60% [25,33], four studies scored 70% [23,29,30,36], five studies scored 80% [21,24,27,28,37], six studies scored 90% [22,26,32,34,35,39], and one study achieved a full score of 100% [38] (Table 5).

### 3.8. Synthesis of the Reports

The external search identified 15 governmental reports from developed and emerging health systems. Among these reports, only two reports (One from Brazil and one from Southeast Asia) were published in peer-reviewed journals [39,40]. The remaining reports were described and published on the local website for the country’s health authority or MoH. Saudi Arabia has the highest governmental report to guide and regulate TH practice. The total published reports were 5 from different authorities, such as the MoH [41,42,43], Saudi Food and Drug Authority [44], and the Council of Health Insurance [45]. The WHO was the second-highest authority to publish four documents related to TH legalization [46,47,48,49], Figure 5. A summary of the grey literature appraisal is provided in Appendix A.

#### 3.8.1. World Health Organization

The Reports from WHO mostly provide advisory guides to help local and international authorities plan TH implementation in their systems. The report provides different definitions for TH, and it was mostly general and not specific to developed or emerging countries. The reports also recommend and suggest global standards and policies for health systems to initiate regulatory guidelines to use TH, as well as encourage health systems to invest in TH solutions, particularly the reports that were published in 2012 [46,48]. The reports published in 2020 provide more technical advice on designing regulatory guidelines that consider people with disabilities who use TH on a daily basis [47]. Additionally, the report published in 2022 provides more recommendations on TH licenses and prescribing medications via TH applications [49].

#### 3.8.2. Saudi Arabia

Different health authorities issued the report from Saudi Arabia. The report by the MoH in Saudi Arabia provides guidelines and details on how TH is regulated in healthcare practice. The report defined TH and other terms referring to TH in the health system. Also, it clarifies a statement regarding using TH as *“health practitioner may conduct examinations or treatment through the services of TH in homes and workplaces, following the regulations set by the MoH.”* Furthermore, there were some guides for the practice on how to obtain a license in TH, as all clinical facilities were mandated to have TH services for patients [41]. In 2021, the National Health Information Centre published the regulatory guidelines for using TH accompanying the services, as well as the stakeholders who participated in the guidelines. For example, the guidelines demonstrated the scope of using TH for screening patients, monitoring patients, consultations, and support provision of the treatment plan [41,43]. In 2022, the legal regulations for using TH were published to set standards, policies, and guidelines for using TH, and to monitor compliance of health authorities with these standards [43]. The document explains the insurance coverage as well as the liability of my malpractice for TH.

Based on the reports by the MoH, in 2023, the Council of Health Insurance [CHI] in Saudi Arabia published a document to provide the requirements for a clinical setting to be approved by CHI to provide TH services and the insurance coverages for the services for both public and private sectors. The report always regulated the conditions related to TH devices, healthcare practitioners, licensing, and data sharing policy [43]. Additionally, the Saudi Food and Drug Authority published a guideline for the TH devices approval process. The advisory guide demonstrates the step-by-step process from approving and marketing TH devices [44].

#### 3.8.3. United Kingdom

In the United Kingdom, the guidelines published by the National Health Service (NHS) demonstrated the TH regulations in England, Scotland, Wales, and Northern Ireland. The report demonstrated that the process of using TH services by the healthcare practitioner must be regulated by the General Medical Council. Additionally, the regulatory guidelines for each healthcare country within the UK are different based on demand. The report also defined TH applications, for example, considering software as TH applications based on medical device regulations in the UK. Furthermore, the document introduced how health authorities deal with and protect data generated by or from TH solutions [50].

#### 3.8.4. United States

In the United States, the report reviewed TH policies in all states. The primary objective of the report was to enhance healthcare access through TH. The report rated the states based on the level of extent the state aligned with the best practices as well as recommendations for optimizing and sustaining TH [51].

#### 3.8.5. Indonesia, Thailand, and Vietnam

The reports from Indonesia, Thailand, and Vietnam were summarized in one article review. The regulatory guidelines mostly discussed a list of definitions related to TH, the infrastructure of TH in clinical practice, licensing TH services in the countries, the tasks and responsibilities of the facilities that provide TH services, and the vision plan to operate TH services in these countries [40].

#### 3.8.6. India

The report from India was issued by the Indian Medical Council Act and the Information Technology Act. These authorities were responsible for setting regulations to govern the practice of TH and ensure its compliance with ethical and legal standards. The guidelines also offer comprehensive details about the technology platforms and tools recommended for efficient healthcare delivery [52].

#### 3.8.7. Brazil

The article maps the regulatory framework of telemedicine in Brazil, highlighting the significant milestones and phases in its development. The study identifies 79 telemedicine-related legislations from the federal government and 31 regulations from federal health professional councils. These regulations are categorized into three historical phases: Formulation/Decision-Making, Organization/Implementation, and Expansion/Maturation [39] (Table 6).

## 4. Discussion

### 4.1. Main Findings

The findings from the literature demonstrate that there is an effort to establish regulatory guidelines for adopting TH in COPD care in several developed countries. This effort varies from one location to another. There are general guidelines on using TH for COPD care in developed and emerging health systems, considering varying degrees of depth in regulatory guidelines and which stakeholders will be impacted by them. There is no theoretical framework for guiding adoption in most of the guidelines. The suggested frameworks explain TH adoption barriers less in depth. Organizations and user-related barriers proved to be the most common, with TH receiving less attention. Current TH regulations are often general and lack COPD-specific provisions. Since some health systems still regulate TH more broadly, comprehensive regulatory guidelines for adopting TH in COPD cannot be achieved. Various government reports from developed and emerging health systems describe TH options, stakeholders impacted by regulatory guidelines, and organizations involved in shaping TH provision. Compared to other regulatory guidelines, Saudi Arabia’s guidance was the most comprehensive. However, the standardization of COPD care regulatory guidelines needs to be improved locally and internationally.

The absence of consistent terminology poses challenges for marketing, evaluating, regulating, and implementing TH in COPD care. It also complicates the synthesis of evidence across studies and may hinder the development of unified guidelines and policies. This highlights a pressing need for consensus on terminology and classification in future research and clinical practice to enhance comparability and ensure coherent regulatory and implementation strategies.

Despite the variation in regulatory maturity across healthcare systems, the presence of regulatory guidelines remains a critical enabler for the sustainable and safe deployment of TH in clinical practice. These regulatory guidelines play an essential role in ensuring patient safety, clinical accountability, and interoperability of digital systems. Nevertheless, the heterogeneity of regulatory approaches across healthcare systems highlights the need for greater harmonization through global or regional frameworks. Such alignment would facilitate broader adoption, enhance implementation scalability, and foster international collaboration in TH applications in COPD care.

In order to effectively manage and monitor COPD patients, it is crucial to have clear and well-structured TH regulations. While TH can significantly enhance COPD care, its adoption and implementation require a deep understanding of the barriers to establishing regulatory guidelines within the health system [54,55]. In general, regulatory guidelines exist but use varying terminology. Most current regulations focus on clinical application, often omitting operational or policy-level considerations [56,57]. The obstacles related to regulatory guidelines arise from various organizational, technological, and stakeholders’ standpoints. Additionally, there are unresolved questions about how to integrate these regulatory guidelines into care models, the responsibilities of healthcare providers, and the benefits for patients. Furthermore, there is a lack of qualitative data that captures stakeholders’ perspectives on the impact of these regulatory guidelines on TH adoption. Stakeholders, individuals or groups, serve as catalysts for change, facilitating the institutionalization of TH through advocacy, leadership, and strategic influence. Their engagement ensures that policies are not only technically sound but also aligned with broader health system goals, making regulatory guidelines more practical, sustainable, and responsive to real-world challenges [58].

Several theoretical models have been proposed to guide the implementation of TH innovations. The Knowledge-to-Action (KTA) framework, for instance, offers a process-oriented view of knowledge translation but primarily focuses on institutional actors and neglects patient-level dynamics [59]. Similarly, the Technology Acceptance Model (TAM) emphasizes user attitudes and perceived ease of use but offers limited insight into systemic or regulatory challenges [60]. More recent frameworks, such as the Tool + Team + Routine model, attempt to bridge the gap between technology, users, and organizational workflows, but lack sufficient detail to explain how regulatory factors influence real-world adoption [34]. Notably, these models often fall short in addressing settings involving multiple stakeholder groups, contextual variations, and policy-level barriers that are especially pronounced in developed healthcare systems [58].

In the case of Saudi Arabia, efforts to scale TH remain hindered by a lack of regulatory clarity. Although Vision 2030 has prioritized digital health transformation as a national goal, the corresponding policies and legal instruments have not evolved at the same pace [15,16]. Many healthcare providers report uncertainty including standards compliance, data protection, and system interoperability when implementing TH for COPD care [61]. Furthermore, there is little empirical evidence examining how healthcare providers, administrators, and patients perceive existing regulations or how they navigate the regulatory guidelines. This disconnect between policy vision and clinical implementation underscores the urgent need for a regulatory framework that is locally grounded, stakeholder-informed, and adaptable to different levels of healthcare infrastructure.

International comparisons reveal similar patterns. In countries such as the UK and USA, structured TH regulations have been instrumental in expanding access and standardizing care [51,62]. Conversely, in countries like Brazil, regulatory saturation where multiple overlapping policies exist without clear implementation roadmaps has hindered widespread adoption [39]. These global experiences underscore the need for a balance between regulatory specificity and operational flexibility, particularly for chronic disease management.

Improved regulatory clarity plays a critical role in shaping patient preference, engagement, and adherence to TH services. When regulatory guidelines are transparent and well-communicated, patients are more likely to trust digital health solutions, particularly in contexts where privacy, safety, and quality of care are clearly protected [58]. Defined protocols around licensure, data security, and accountability reassure patients that the care provided is equivalent to traditional in-person services. Having this clear regulatory guideline often translates into increased patient willingness to adopt and consistently use TH platforms.

Furthermore, regulatory clarity fosters consistency in service delivery, which enhances patient satisfaction and long-term adherence. When patients experience streamlined access to TH without confusion around eligibility, insurance coverage, or technical barriers, they are more engaged and proactive in managing their health. In chronic disease care, such as for COPD, this clarity directly contributes to improved health outcomes, as patients are more likely to attend virtual follow-ups, comply with treatment plans, and communicate regularly with care teams. Overall, the alignment of regulatory policies with patient-centered care principles is essential to advancing the sustainable integration of TH in healthcare systems.

Addtionally, variation in studies quality underscores the importance of critically evaluating the evidence base used to inform TH regulatory guidelines. High-quality studies are essential for developing effective, contextually appropriate, and sustainable policies in COPD care.

### 4.2. Proposed Framework to Succeed TH Implementation

To address the multifaceted challenges associated with the adoption of TH, we propose a simple, novel framework aimed at enhancing the successful implementation of TH initiatives. As illustrated in Figure 6, the framework emphasizes the integration of key operational and strategic elements, including the involvement of end-users, stakeholders, and local business partners, as well as the tailoring of innovations to specific contexts. It also incorporates essential factors such as interdisciplinary collaboration, feedback mechanisms, and the availability of feasible business models. Additionally, it recognizes the importance of adapting to organizational changes and anticipating system-level transformations required for long-term integration. This framework provides a practical guide for policymakers, implementers, and healthcare organizations to structure their TH strategies effectively.

### 4.3. Strengths and Limitations

This systematic review offers a comprehensive synthesis of global and regional literature on regulatory guidelines influencing TH adoption in COPD care. A key strength lies in the dual inclusion of peer-reviewed studies and grey literature, enhancing contextual relevance and policy insights. The use of the CASP tool ensures methodological rigor and appraisal consistency across qualitative studies. Additionally, the review provides a structured categorization of barriers and facilitators, supporting informed policymaking.

However, several limitations should be acknowledged. First, the lack of studies specific to Saudi Arabia (all were governmental reports) limited the depth of region-specific analysis. Second, the exclusion of non-English sources may have overlooked valuable regional reports. Third, the heterogeneity in terminology and frameworks across studies posed challenges in direct comparison. Fourth, current regulatory guidelines reports adopted by health authorities tend to emphasize general operational and technical standards, offering limited guidance for disease-specific applications such as COPD management. Finally, while narrative synthesis was appropriate, the absence of meta-analytic techniques limits generalizability. More research is needed to evaluate the impact of the current regulatory guidelines on the market adoption and implementation of TH in COPD care locally and internationally.

## 5. Conclusions

This review highlights the critical role of regulatory guidelines in shaping the adoption and implementation of TH solutions for COPD care. Despite global progress, the evidence suggests that existing regulations often lack specificity, stakeholder engagement, and contextual adaptation. Saudi Arabia has shown commendable progress in digital health policy under Vision 2030; however, TH guidelines remain fragmented and insufficiently tailored for chronic disease contexts such as COPD.

To promote effective TH adoption in COPD care, regulatory guidelines must move beyond generic mandates and embrace frameworks that address disease-specific, stakeholder-driven, and system-level variables. Future regulatory efforts should incorporate integrated models and frameworks, adapted to reflect regulatory interactions and policy barriers from different directions. Policymakers should also prioritize clear responsibilities, data governance, and cross-platform interoperability to bridge the gap between digital health ambitions and practical implementation. These guidelines must be co-developed with input from clinicians, patients, administrators, and technology vendors to ensure alignment with real-world clinical routines and resource constraints.

Promoting regulatory clarity begins with policymaking that centers the experiences and perspectives of all stakeholders. This includes conducting qualitative studies to capture how regulations affect provider workflows, patient trust, and organizational readiness.

Ultimately, our review calls for the development of practical, inclusive, and evidence-informed regulatory frameworks that accommodate both clinical and technological realities. Future research should prioritize qualitative studies to assess the barriers and facilitators toward TH implementation.

## Figures and Tables

**Figure 1 healthcare-13-02858-f001:**
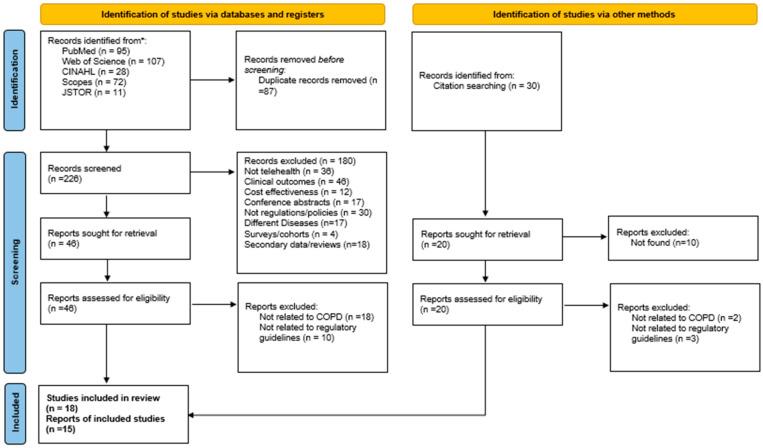
PRISMA flow diagram. * means databases.

**Figure 2 healthcare-13-02858-f002:**
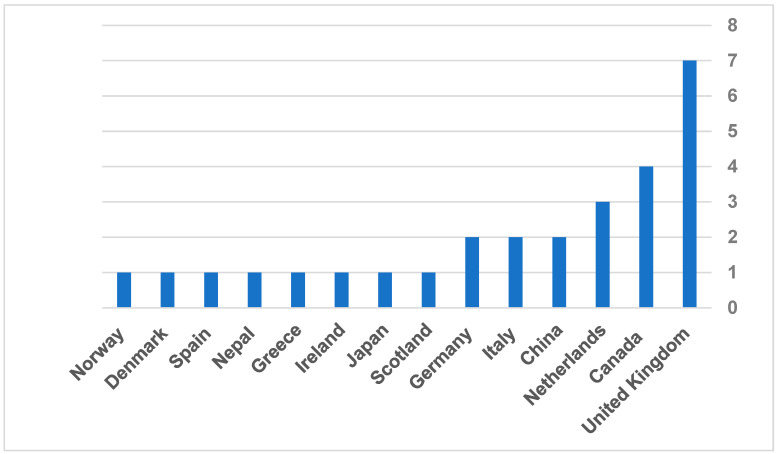
Frequency of countries included in the published articles.

**Figure 3 healthcare-13-02858-f003:**
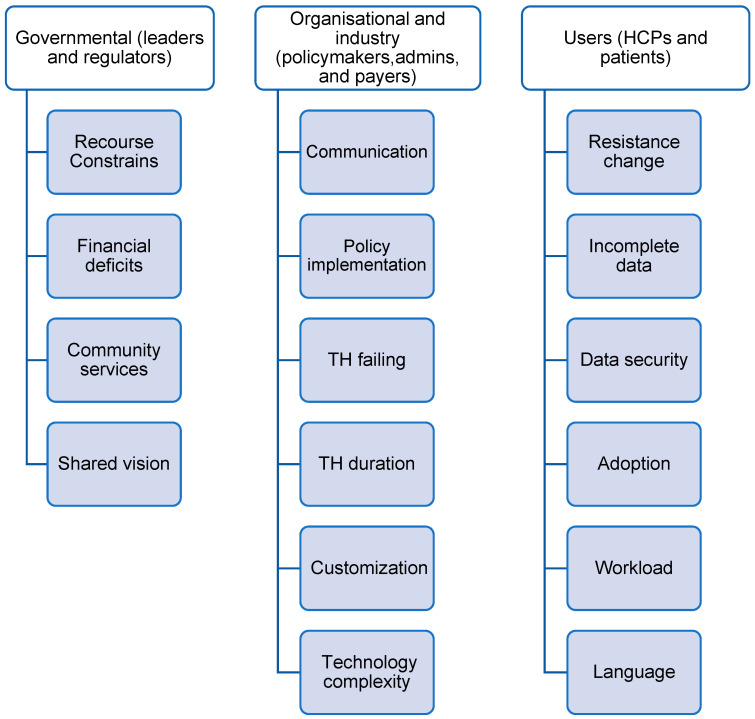
Barriers identified from included studies.

**Figure 4 healthcare-13-02858-f004:**
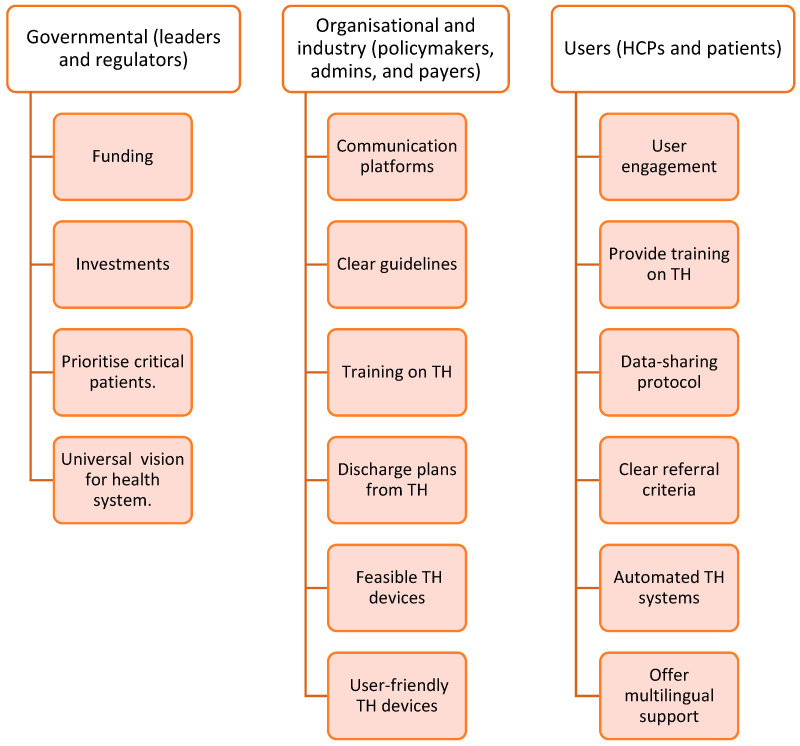
Facilitators identified from the included studies.

**Figure 5 healthcare-13-02858-f005:**
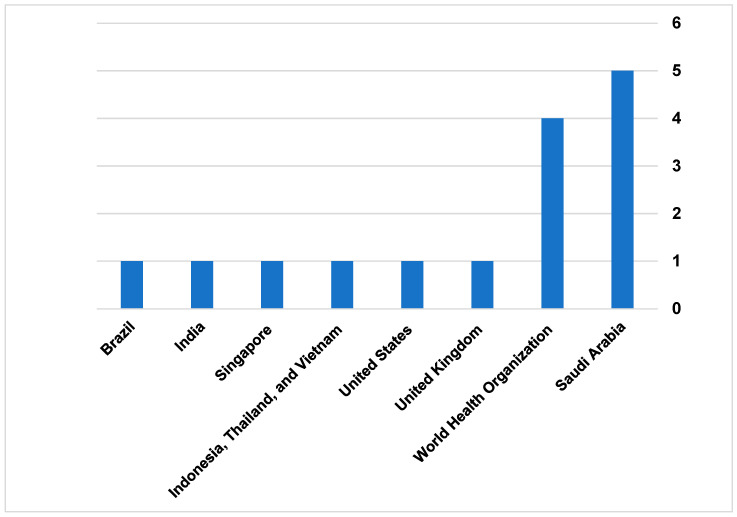
Countries of governmental reports.

**Figure 6 healthcare-13-02858-f006:**
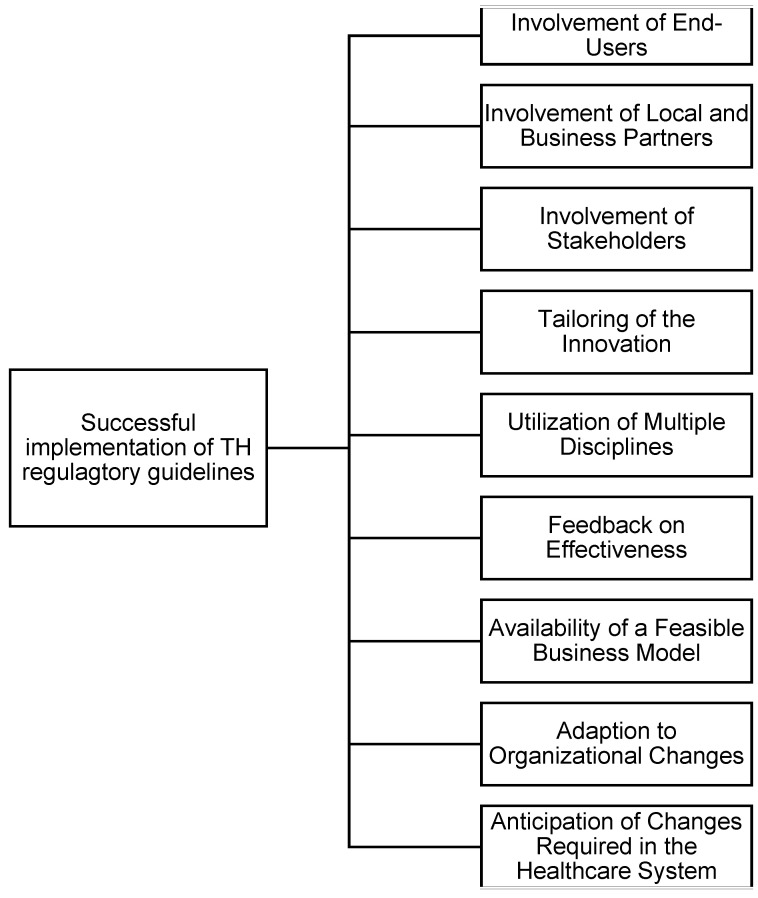
Factors impacting the successful implementation of TH (Created by author).

**Table 1 healthcare-13-02858-t001:** Search Strategy.

Theme	Key Terms [MeSH Terms] ^a^
Chronic Obstructive Pulmonary Disease	pulmonary disease, chronic obstructive [MeSH Terms] OR “chronic obstructive pulmonary disease” OR “COPD”
“AND”	
Regulatory Guidelines	“Regulatory” OR “guideline” OR “guidelines as topic” OR “guidelines” OR “law” OR “regulation” OR “legal” OR “legislation” or “regulat” or “policy” or “policies” OR “Legislation” OR “Legislation, Medical”
“AND”	
Telehealth	“telehealth” OR “tele-health” OR “biomedical device” OR “digital health technolog” OR “mhealth” OR “m-health” OR Biomedical Technology OR telemedicine

Footnote: ^a^: MeSH; Medical Subject Terms. Search initially performed in March 2025, updated in October 2025.

**Table 2 healthcare-13-02858-t002:** Inclusion and exclusion criteria of included studies.

Action	Eligibility Criteria
Included	Studies must explore the facilitators and barriers to TH regulatory guidelines implementation in COPD care.
Included	The search prioritized qualitative studies that investigated the barriers and facilitators of TH implementation in COPD care.
Included	Studies must evaluate the compliance of TH solutions with regulatory standards and policies.
Included	Studies must investigate stakeholders’ and users’ perspectives on regulatory guidelines for TH applications in COPD care or related contexts.
Included	When possible, studies must present or introduce a theoretical framework that explains regulatory guidelines implementation. Proposed theoretical frameworks to guide future regulatory guidelines for TH in COPD were included as well.
Included	Included studies must be original and present primary findings.
Included	Included studies must be available in full text.
Included	For inclusion of reports, it must be specific for regulatory guidelines for TH implementation. General reports related to regulatory guidelines for TH implementation and development have been included.
Included	Only studies and reports written in English were included.
Excluded	Studies focusing on diseases other than COPD were excluded.
Excluded	Studies that examined public perceptions of TH applications through surveys or questionnaires were excluded.
Excluded	Reviews and secondary data analysis papers were excluded.
Excluded	Conference abstracts and summary articles were excluded.

**Table 3 healthcare-13-02858-t003:** Summary of included studies.

Author(s)	Country	Year	Summary of the Findings About Regulatory Guidelines
Hamilton et al. [21]	UK	2008	There is a need to create a regulatory guideline for TH use in primary care settings for COPD patients. The main driver for creating regulatory guidelines is the need to enhance care delivery for long-term conditions by transitioning it into community-based settings. The study identified barriers to creating regulatory guidelines, such as local influences and financial constraints that impact service design and implementation. Policymakers and administrators have acknowledged a significant disparity between policymakers and healthcare practitioners in understanding TH and its value.
Elwyn et al. [22]	UK	2012	The lack of regulatory guidelines for TH services in COPD due to: ○ Lack of positive clinical outcomes that promote the advantages of TH. ○ The existing TH devices are impractical for sustained patient use. ○ This is needed to automate TH services where data is collected and analyzed automatically. ○ Interprofessional relationships and legislation during the use of TH services are not adequately described in the general guidelines, which makes it difficult for policymakers to identify the responsibilities for each profession.
Walters et al. [23]	Netherlands	2012	TH policy and regulations for COPD patients faced several barriers, such as ○ Resistance from workers to change the health system. ○ Introducing patient-centered care or tailored care as a health approach. ○ Changing the current technological system and integrating it into the health system. ○ Integrating new TH innovations into a larger system takes more time.
Odeh et al. [24]	UK	2014	There were some challenges that slowed down the implementation of TH services in COPD care The local organizational regulatory guidelines did not specifically describe the use of TH services in COPD care. The general regulatory guidelines did not describe the patients who are suitable for TH services. The lack of resources, including a specific TH team, existence of TH devices in all departments, insufficient training regarding TH devices, and inadequate technical support.
Dirven et al. [25]	Netherlands	2014	Policies and regulations of using TH must not rely on feasibility and effectiveness of TH solutions. Multiple stakeholders must be included in the formulation of policy and regulations of TH.
Taylor et al. [26]	UK	2015	Policymakers agreed that it is hard to identify reliable and flexible TH services, and if they are included in the services, staff would need appropriate training and support. Several factors hinder policymakers from implementing regulatory guidelines for TH and COPD management. These include adapting to a dynamic environment, introducing TH to frontline staff, gaining firsthand experience and understanding of TH, addressing technology and service design challenges, and seamlessly integrating TH into routine care. Early adoption and implementation of TH services must be guided by clear regulatory guidelines, which involve policymakers and healthcare practitioners. Regulatory Guidelines for using TH in COPD faced conflict attitudes, but mostly, there was a resistance from healthcare providers to accept a change in services to TH. Lack of motivation and increased workload were the most factors identified by the frontline healthcare providers.
Hunting et al. [27]	Canada	2015	Regulatory guidelines in adopting TH solutions in COPD care have some barriers and facilitators: Challenges encompassed issues related to access, patient language, provider time constraints, gaps in care delivery, and cultural obstacles. Key facilitators encompassed user-friendly technology, patient motivation, robust provider support, seamless integration into broader health service, and thorough evaluation.
Rojahn et al. [28]	UK, Germany, Italy, Spain	2016	Regulatory guidelines on TH were identified in all four countries for chronic diseases. However, pilot projects existed or were planned for patients with chronic diseases such as COPD. There is a lack of formal identification within the four countries of the current TH devices used in patient care. Initiatives and incentives for using TH are required to increase usability and build evidence that contributes to helping regulators and policymakers in their decisions.
Segato et al. [29]	Italy	2017	Sustainable TH interventions for COPD care need regulations, policies, and legislation: (1) ensuring transparent reimbursement for professionals, (2) providing adequate incentives to encourage implementation and long-term sustainability, and (3) prioritising patient safety and privacy Organizations must initiate new guidelines and train staff, such as nurses and respiratory therapists, on them. There are other factors that impact the regulatory guidelines implementation in an organization, such as: ○ Task responsibilities that must be clear in the regulatory guidelines ○ Technology must be supported by professional teams for technical support. ○ Implementing new workflows and adopting novel communication platforms. ○ Adjusting to alternative modes of communication. ○ Creating incentives to foster collaboration between hospital and primary care structures while maintaining transparent communication. ○ Ensuring sufficient funding for TH procurement, maintenance, and ongoing operational costs. ○ Developing an exit strategy to sustain the service beyond the initial funding period. ○ Providing training for both patients and professionals on TH usage.
Keene et al. [30]	UK	2019	The study investigated the perspectives of various stakeholders involved in regulating TH solutions in clinical studies such as patients, physicians, regulators, and payers. To regulate any TH solutions for COPD patients: ○ Under health technology assessment, payers (such as insurance companies or government agencies) consider the value and cost-effectiveness of treatments. ○ The treatment policy must have different options of devices for the patients. ○ The treatment policy usually estimates treatment effects based on how patients take the treatment. ○ It is better for the TH policy to be initiated as a “while-on-treatment” adjunct solution. ○ TH must be a tool which measures the treatment efficacy and communication.
Slevin et al. [31]	Ireland	2020	Regulating TH in COPD care and implementing TH services faced the following challenges: ○ Ensuring data quality, implementing evidence-based care, addressing resource limitations, and promoting digital literacy. Possible solutions included enhancing digital health training and education, bolstering HCP’ digital literacy, tailoring prescriptions to individual needs. And adopting patient-centered approaches (such as pulmonary rehabilitation and shared decision-making).
Gaveikaite et al. [32]	Greece Denmark, Norway, Netherlands, Scotland, Italy	2020	Regulatory guidelines to adopt TH solutions explored by nurses and physicians as follows: Nurses play a pivotal role in the adoption process, and change management should prioritize addressing their needs. Presence of change management leads to immediate TH adoption. Nurses’ ability to make independent decisions. Without workload increase, change management is unlikely to start. Nurse must be aware of the diseases as well as the history of the patients. Physicians face the most complex adoption process, where perceived value is critical. Change management and holistic patient understanding are important factors that indirectly influence regulatory guidelines. Centralization of services negatively impacts the Physician-Patient Relationship Champion presence leads to selective activation of staff, which increases the chances of Routinization. This, in turn, can enhance Adoption among initially hesitant physicians
Alwashmi et al. [33]	Canada	2020	Policy and regulation makers must consider the following factors when planning development of a TH intervention in COPD care: Patient interface must consider patient education, gather baseline and subjective data, collect objective data using compatible medical devices, offer a digital action plan, enable progress tracking, allow family access, customize features, remind patients, and reward positive behaviors. HCP interface features must consider tracking patient progress, communication, education, and rewards.
van Lieshout et al. [34]	Canada	2020	To design a policy and regulations for a service that includes TH and innovation, the adoption must include (1) Evaluation of a unique value proposition of a specific tool. (2) Assessment of how the tool impacts team dynamics and relationships (3) Clear definition of the normal routines necessary for integrating technology into the service.
Yadav et al. [35]	Nepal	2021	- Integrated models of care that include TH solutions must be regulated by multiple stakeholders. Existence of model of care in a health system can be used to guide policy and regulations for TH innovations in COPD care. Implementation and adoption of TH innovations in policy and regulations influenced by - Internal factors such as motivation to change, familiarity with technology, meeting needs, existing gaps, respect and trust, sense of equality, benefits, value, learning new things. - External factors such as attractive and comfortable environment, understanding of content, reimbursement, engagement. - Other factors such as contextual creativity, principles of co-design, tools and process.
Qingfan et al. [36]	China	2021	It is essential to establish a framework to inform regulatory and policy decisions regarding TH which is based on Direct Patient Care Regulatory and Policy Compliance Digital Health Intervention Research and Design.
Anne- Meiwald et al. [37]	Japan, Canada, England, Germany	2022	Common barriers toward regulatory guidelines of TH in COPD care ○ Health systems in these countries faced advanced COPD populations. There is low consideration of COPD care in the health systems. - Acute and chronic COPD patients received sub-optimal disease management because of the low consideration of TH solutions. - COVID-19 was the drive to increase attention to start regulating using TH solutions in COPD care. - Regulatory guidelines must take into consideration the previous policy and guidelines regarding the diagnosis of COPD, the treatment options, and the impact of COVID-19 on disease.
Yuyu et al. [38]	China	2022	Regulatory guidelines in using TH are influenced by the following factors: (1) Faced with a vast amount of online health information (Professional). (2) Essential competencies and personality traits ensuring older patients’ participation and sustained use (Patients) (3) User experience with technology (Organization/industrial) (4) Being in a complex regulatory context (Policymakers)

**Table 4 healthcare-13-02858-t004:** Summary of the theoretical frameworks.

Author (Year)	Framework Name	Strengths	Limitations	Potential to Endorse Regulatory Guidelines
Iqbal et al. (2022) [13]	Traditional Industry-Structure Framework	Addresses regulatory gaps; emphasizes adaptive frameworks, public–private partnerships, and international cooperation.	Lacks specific implementation details; may be complex to operationalize in diverse systems.	Can guide dynamic and collaborative regulatory models that evolve with technology.
Hunting et al. (2015) [27]	Multi-Level TH Framework	Analyzes factors across multiple domains; highlights facilitators and barriers to adoption.	Context-specific to Ontario; structural challenges may limit generalizability.	Supports comprehensive evaluation of systemic and contextual barriers to inform regulations.
Gaveikaite et al. (2020) [32]	Causal Loop Diagram (CLD)	Reveals dynamic feedback loops; highlights perceived value and change management needs.	Focused on physicians; limited inclusion of broader healthcare actors.	It can inform behavior-driven regulatory design tailored to stakeholder motivations.
Yadav et al. (2021) [35]	Integrated Model of Care	Emphasizes stakeholder engagement and community involvement.	Logistical and cultural challenges; limited scalability.	Useful for community-based regulatory frameworks in emerging health systems.
Qingfan et al. (2022) [36]	Digital Health Interventions (DHI) Stakeholder Map	Maps internal and external stakeholders; supports health information exchange.	Abstract model; lacks implementation specifics.	Can assist in defining roles and responsibilities within telehealth governance structures.
Segato and Masella (2017) [29]	Italian Framework	Identifies key factors: technology, acceptance, organizations, financing, policy and legislation.	May lack flexibility; context-specific to Italy.	Provides a structured basis for designing sustainable TH policies.

**Table 5 healthcare-13-02858-t005:** Quality of studies.

Study	Q1	Q2	Q3	Q4	Q5	Q6	Q7	Q8	Q9	Q10	%Positive
Alwashmi et al. [33]	yes	yes	can’t tell	yes	can’t tell	yes	yes	can’t tell	can’t tell	yes	60%
Anne Meiwald et al. [37]	yes	yes	no	yes	yes	no	yes	yes	yes	yes	80%
Dirven et al. [25]	yes	yes	no	yes	can’t tell	no	can’t tell	yes	yes	yes	60%
Gaveikaite et al. [32]	yes	yes	no	yes	yes	yes	yes	yes	yes	yes	90%
Elwyn et al. [22]	yes	yes	yes	yes	yes	can’t tell	yes	yes	yes	yes	90%
Hamilton et al. [21]	yes	yes	yes	yes	yes	can’t tell	no	yes	yes	yes	80%
Hunting et al. [27]	yes	yes	can’t tell	yes	yes	no	yes	yes	yes	yes	80%
Keene et al. [30]	yes	yes	no	yes	yes	yes	no	no	yes	yes	70%
Odeh, et al. [24]	yes	yes	can’t tell	yes	yes	yes	no	yes	yes	yes	80%
Qingfan et al. [36]	yes	yes	yes	yes	yes	can’t tell	no	no	yes	yes	70%
Rojahn et al. [28]	yes	yes	yes	yes	yes	yes	no	can’t tell	yes	yes	80%
Segato et al. [29]	yes	yes	can’t tell	can’t tell	yes	yes	no	yes	yes	yes	70%
Slevin et al. [31]	yes	yes	yes	yes	yes	yes	no	yes	yes	yes	90%
Taylor et al. [26]	yes	yes	yes	yes	yes	can’t tell	yes	yes	yes	yes	90%
van Lieshout et al. [34]	yes	yes	can’t tell	yes	yes	yes	yes	yes	yes	yes	90%
Walters et al. [23]	yes	can’t tell	can’t tell	yes	yes	yes	no	yes	yes	yes	70%
Yadav et al. [35]	yes	yes	no	yes	yes	yes	yes	yes	yes	yes	90%
Yuyu et al. [38]	yes	yes	yes	yes	yes	yes	yes	yes	yes	yes	100%

**Table 6 healthcare-13-02858-t006:** Summary of governmental and organizational reports.

N	Country/Type of Document	Year	Regulation/Legislation Summary
1	WHO (Global) [46]	2012	The “Digital Implementation Investment Guide (DIIG)” is a guide developed by the World Health Organization. The objective is to assist governments and technical partners in devising a digital health strategy that prioritizes specific health programs, aligning with national health system objectives for the implementation of TH services. The guide outlines a structured approach for countries to create a budgeted implementation strategy for digital health across one or more health systems. The guide offers recommendations based on WHO-recommended digital health interventions that align with national digital architecture, country readiness, health system priorities, and policy objectives.
2	World Health Organization (Global) [48]	2012	The report is a strategy-building guide for health systems to start regulating TH. The toolkit offers practical guidance for creating and implementing national eHealth visions, action plans, and monitoring frameworks. It serves as a valuable resource for governments, ministries, and stakeholders, regardless of a country’s level of development. By leveraging eHealth strategies, healthcare systems can become more efficient and responsive to people’s needs.
3	WHO (Global) [47]	2020	The document titled “WHO-ITU Global Standard for the Accessibility of TH Services” provides technical requirements to ensure TH platforms are accessible with TH services for all The standard emphasizes the compatibility of TH platforms with assistive devices, the provision of captioning and text messaging, the avoidance of background music in videos, and the inclusion of clear subtitles. The report updated TH policy.
4	Brazil (Review paper) [39]	2020	The legal framework for telemedicine in Brazil has evolved over the past 30 years, with significant developments occurring since 2011. The Federal Council of Medicine was the most active in standardizing TH, responsible for 67.7% of the regulations. Despite the proliferation of legislation and regulations, there is still no fully consolidated regulatory framework for TH in Brazil. The Federal Nursing Council prohibits nurses from complying with remote medical prescriptions without a doctor’s stamp and signature, but an exception is made for TH medical prescriptions. Key themes include the establishment of TH programs, the integration of telemedicine into the Brazilian Unified Health System (SUS), the development of evaluation tools for e-health services, and the financial incentives for TH centers.
5	Saudi Arabia (MoH) [41]	2021	The legal framework for TH in Saudi Arabia comprises the TH Application Guidelines, the Governing Rules of TH in Saudi Arabia, and the regulations of the National Health Information Centre. These regulations work together to govern the practice of TH and ensure its adherence to ethical and legal standards. The guidelines also include comprehensive details on the technology platforms and tools to be used for effective healthcare delivery.
6	Saudi Arabia (Food and Drug Authority) [53]	2021	The report regulates the use of TH services and medical devices for safety, market, and adoption. Also, it summarizes the process of getting TH license for a medical device.
7a	Indonesia (Ministry of Health) [40]	2021	The report was published in local language, but it summarizes the list of definitions related to TH, the infrastructure of TH in clinical practice, licensing TH services in the country, and tasks and responsibilities of the facilities that provide TH services. Clinical governance, cost, reimbursement and funding are also described in the report.
7b	Thailand (Ministry of Public Health) [40]	2021	The report defined ICT related to health and the benefits of using TH. Also, the report discusses the infrastructure and readiness of the health system for adopting 5 5-year action plan to comply with the system.
7c	Vietnam (Ministry of Health) [40]	2021	The report defined TH as distant medicine. The report discusses technical aspects such as confidentiality and data privacy related to licensing TH services
8	Singapore (Ministry of Health) [42]	2021	The Health Service Authorities (HAS) issue guidelines to clarify which types of TH products fall under medical device regulation. These products encompass hardware devices, software, and mobile applications. Whether a TH product is regulated as a medical device depends on its intended use. TH products designed for medical purposes—such as investigation, detection, diagnosis, monitoring, treatment, or management of medical conditions, diseases, anatomy, or physiological processes—are classified as medical devices subject to regulatory controls by HSA. TH products designed for wellness, such as fitness trackers that can also perform medical functions like heart rate monitoring, must include a “clarification statement” on their labels to inform users of the appropriate use. TH medical devices are classified into different risk classifications depending on the nature of the device and its intended functions. The company is required to obtain marketing clearance for the device from HSA via Product Registration before supply of the devices in Singapore, unless the device is a Class A device. Class A TH medical devices are exempt from Product Registration with HSA. Dealers of TH medical devices must fulfill post-market responsibilities, such as reporting adverse events, defects, and recalls to the HSA.
9	Indian (Medical Council of India) [52]	2021	India’s telemedicine legal framework includes the TH Practice Guidelines, the Indian Medical Council Act, and the Information Technology Act. These regulations collectively oversee telemedicine practices, ensuring they comply with ethical and legal standards. Additionally, the guidelines provide comprehensive details on the technology platforms and tools necessary for effective healthcare delivery.
10	WHO(Global) [49]	2022	Resources on eligible services, providers, sites, and more are available. Healthcare providers eligible for Medicare billing can bill for TH services, regardless of patient or provider location. TH licensure requirements differ at the federal, state, and cross-state levels for healthcare providers. Authorized providers are able to prescribe controlled substances via TH if they meet certain criteria. Recent Federal legislation and policies related to TH are accessible.
11	United Kingdom (NHS) [50]	2022	Various regulatory bodies specific to each country oversee TH services in the UK: the Care Quality Commission (CQC) in England, Healthcare Improvement Scotland (HIS), Healthcare Inspectorate Wales (HIW), and the Regulation and Quality Improvement Authority (RQIA) in Northern Ireland. Individual medical practitioners involved in TH practice fall under the regulation of the General Medical Council (GMC). All doctors practicing medicine in the UK must be GMC-registered and adhere to specific standards for good medical practice. Regulatory bodies in each country have issued guidance to the professionals on how to use TH. CQC has proposed regulations for digital healthcare providers in primary care. The software utilized in TH may qualify as a medical device under the Medical Devices Regulations 2002 (UK MDR). Much of the data collected through TH falls under the category of ‘data concerning health’ according to the UK General Data Protection Regulation, making it ‘special category data.’ The UK government is likely to introduce additional legislative measures to integrate remote healthcare provision into public healthcare goals.
12	Saudi Arabia (MoH) [43]	2022	The Royal Order (No. 47455) issued on 9/8/1441 AH provides an exception to Article 13 of the Law of Practicing Health Professions (Royal Decree No. M/59 on 4/11/1426 AH), allowing health practitioners to perform examinations or treatments via TH services in homes and workplaces, as per the regulations established by the MoH. Only healthcare facilities licensed by the MoH are permitted to provide TH services. All healthcare facilities should offer TH services to provide care. Any entity offering TH services without being licensed as a healthcare facility in the Kingdom of Saudi Arabia must obtain a license from the TH Centre to practice TH. The practice of TH shall be restricted to the healthcare professionals who are credentialed to practice within the jurisdiction of the Kingdom of Saudi Arabia. Provision shall be made for the health insurance coverage to include TH services. In TH services involving multiple healthcare professionals, each professional is accountable for their own contributions. The TH Centre shall support, monitor, and evaluate the implementation and development of TH services in the Kingdom of Saudi Arabia. Healthcare facilities must ensure a secure and private workspace to protect patient privacy, including personal information, communications, and consulting spaces. Patients have the right to decline or withdraw from TH services at any time without needing to provide a reason.
13	Saudi Arabia (Council Health Insurance) [52]	2023	The report delineates TH policies and procedures overseen by the Council of Health Insurance (CHI). It covers the Saudi billing system, methods of delivering insured TH services, and the prerequisites for healthcare providers seeking CHI accreditation to practice TH. The report includes definitions, conditions for practicing TH, insurance coverage, accreditation requirements, rights of beneficiaries, and dispute resolution. The policy is derived from a Royal Decree amending the Health Professions Practice Law to allow TH services. The document outlines requirements for registration, licensing, preparedness, technical specifications, and qualifications for health practitioners. Additionally, it provides specifics on insurance-covered services, exemptions, pricing, and coverage limits The report highlights the significance of privacy, security, and adherence to Saudi laws and regulations when delivering TH services. It elaborates on beneficiary rights, dispute resolution, insurance-covered TH services (including teleconsultation, telediagnosis, telemonitoring, tele-pharmacy, and remote therapy), and the associated coverage terms. Furthermore, it highlights the significance of routine medical device checks, professional training, high-quality audio-visual equipment, automated data collection, and patient safety. It mentions the financial fee for accreditation, the need for insurance company approval for remote monitoring services, and the beneficiary’s right to cancel or stop services. Additionally, it highlights the Council’s role in reviewing the list of permitted services as directed by the MoH.
14	Saudi Arabia (Food and Drug Authority) [44]	2023	The report regulates paying for TH devices for clinical purposes. Classification of TH services to regulate the permission of use in the clinics.
15	USA (Cicero Institute for Health) [51]	2024	The report assesses TH policies across all 50 U.S. states, aiming to enhance healthcare access through TH. Each state receives a stoplight rating (green, yellow, or red) based on policy alignment with best practices. The report provides actionable recommendations for optimizing TH services for both providers and patients.

## Data Availability

The original contributions presented in this study are included in the article.

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
