# Peer review of "Evaluating the Impact of Regulatory Guidelines on Market Adoption and Implementation of Telehealth for COPD Patients: A Systematic Literature Review"

_healthcare, 2025, doi:10.3390/healthcare13222858_

Round 1
Reviewer 1 Report
Comments and Suggestions for Authors
- Register and cite the systematic review protocol for transparency.
- Include details about how grey literature was appraised for quality and relevance.
- State why meta-analysis wasn’t feasible (e.g., heterogeneity, qualitative design).
- Fix all missing figure references and ensure figures are correctly embedded, especially 3,4,5, and 6 figures are not cited in the text.
- Use content analysis or frequency tables to quantify barrier/facilitator occurrences.
- Perform comparative analysis between high-income and low-income systems using thematic synthesis.
- Reduce repetition and consolidate discussion of CASP in a more concise paragraph.
- p.14 (line 294), p.17 (line 324), p.25 (line 475) - Correct all reference errors and re-check figure/table callouts.
- Strengthen rationale by defining specific knowledge gaps.
Recheck all the English grammar mistakes
Author Response
Reviewer 1
- Register and cite the systematic review protocol for transparency.
- Thank you for your comments. Our review has been registered on the PROSPERO Registry (ID: CRD420251178887).
2. Include details about how grey literature was appraised for quality and relevance.
- Thank you for your comments. We have added details regarding the appraisal of grey literature in the Methods section. The revised manuscript reads: “In addition, grey literature was evaluated using the Authority, Accuracy, Coverage, Objectivity, Date, and Significance (AACODS) checklist.” (Page 4, Lines 131–133).
- State why meta-analysis wasn’t feasible (e.g., heterogeneity, qualitative design).
- Thank you for your valuable comments. We have added this clarification in the revised manuscript: “A meta-analysis was not conducted, as the included studies were qualitative in nature and exhibited heterogeneity in design and outcome reporting.” (Page 5, Lines 139–141).
3. Fix all missing figure references and ensure figures are correctly embedded, especially 3,4,5, and 6 figures are not cited in the text.
- We apologize for this error. The issue has now been corrected. The cross-reference function has been updated and verified. The revised manuscript reads: “A list of 16 barriers and 16 facilitators have been identified from the included studies described in Figure 3 and Figure 4.” (Page 14, Lines 297–299; Page 17, Line 327; Page 25, Line 478).
4. Use content analysis or frequency tables to quantify barrier/facilitator occurrences.
- Thank you for your valuable comments. We have incorporated content analysis and added frequency tables summarizing the barriers and facilitators, presented in Supplementary Tables S2 and S3 as requested
5. Perform comparative analysis between high-income and low-income systems using thematic synthesis.
- thank you. Comparative thematic analysis has been added to the Results section and presented in Supplementary Tables S4 and S5.
6. Reduce repetition and consolidate discussion of CASP in a more concise paragraph.
- Thank you for your comments. We revised the text under methods and results.
- In Methods: It readers “The CASP tool is widely recognized in evidence-based research for its structured approach to evaluating the validity, results, and relevance of qualitative and quantitative studies.[18,19] This tool was chosen due to its comprehensive nature, covering key domains such as methodological rigor, validity of findings, and relevance to the research question. In addition, grey literature was evaluated using Authority, Accuracy, Coverage, Objectivity, Date, and Significance (AACODS) checklist” Page 4 Lines 133-140
- In the results: it reads “The methodological quality of the included studies was assessed using the CASP checklist. Overall appraisal scores ranged from 60% to 100%, indicating moderate to high methodological rigor. Two studies scored 60%, [25,33]four studies scored 70%, [23,29,30,36] five studies scored 80%, [21,24,27,28,37] six studies scored 90%, [22,26,32,34,35,39] and one study achieved a full score of 100%.[38] (Table 5).” Page 15 Lines 300-305
7. p.14 (line 294), p.17 (line 324), p.25 (line 475) - Correct all reference errors and re-check figure/table callouts.
- We apologize for these typographical errors. All reference and figure callouts have been reviewed and corrected in the revised manuscript.
8. Strengthen rationale by defining specific knowledge gaps.
- Thank you for this insightful comment. We have strengthened the rationale by revising the Introduction to focus more clearly on the identified knowledge gaps. The revised manuscript now reads:
- “However, the literature highlights significant gaps in knowledge, practical, and culturally appropriate regulatory frameworks tailored to TH use in COPD care, particularly in developing and emerging health systems. [13–16] Existing regulatory guidelines are generic and lack disease-specific direction. Empirical research on real-world implementation of regulatory guidelines remains limited, and the influence of infrastructural and organizational contexts is unclear. Moreover, no consensus exists on the most suitable theoretical framework to guide TH implementation in COPD care.” (Page 2, Lines 78–84).
Reviewer 2 Report
Comments and Suggestions for Authors
Thank you for the opportunity to review this manuscript. Please find my suggestions and comments.
Subject
Title: “Evaluating the Impact of Regulatory Guidelines on market adoption and Implementation of Telehealth for COPD Patients: A Systematic Literature Review” /Abstract: Conclusion: “Clear, inclusive, and context- sensitive regulatory guidelines are essential to support the successful integration of TH in COPD care” / Lines 399-400: “The findings from the literature demonstrate that there is an effort to establish regulatory guidelines for adopting TH in COPD care in several developed countries”. However, the authors included 15 reports not focused on COPD patients. Please, think about this.
Abstract
Please, describe the date of the study.
Material and Methods
-Line 96: “Examples of search strategies from all databases…” OR “Search strategies from all databases…” Considering a systematic revision, the authors have to write all search strategies, not only “examples”. Please, think about this.
-Lines 106-108: “Therefore, the scope of our electronic search was expanded to encompass governmental and non-governmental reports that discuss regulatory guidelines for TH in COPD care or similar contexts. Which similar contexts? I did not find information about this in “reports” included. I found reports about general telehealth. Please think about this.
-Line 115: STUDIES “…particularly in the context of COPD…” or “…in the context of COPD…” ?
- I did not find who (the authors) and how the authors identified the studies and reports (They independently evaluated the abstracts....). Considering a systematic revision, the authors have to describe the details about this.
Results
-Table 2. Elwyn et al : “…We need more automated…” OR “There is need to …” ? Please, think about this.
-Lines 187-192: “The absence of consistent terminology poses challenges for marketing, evaluating, regulating, and implementing TH in COPD care. It also complicates the synthesis of evidence across studies and may hinder the development of unified guidelines and policies. This highlights a pressing need for consensus on terminology and classification in future research and clinical practice to enhance comparability and ensure coherent regulatory and implementation strategies.” Please, this information could be moved to “Discussion” section.
-Lines 210-217: “Despite the variation in regulatory maturity across healthcare systems, the presence of regulatory guidelines remains a critical enabling for the sustainable and safe deployment of TH in clinical practice. These regulatory guidelines play an essential role in ensuring patient safety, clinical accountability, and interoperability of digital systems. Nevertheless, the heterogeneity of regulatory approaches across healthcare systems highlights the need for greater harmonization through global or regional frameworks. Such alignment would facilitate broader adoption, enhance implementation scalability, and foster international collaboration in TH applications in COPD care.” Please, this information could be moved to “Discussion” section.
-Lines 246-251: “These individuals or groups serve as catalysts for change, facilitating the institutionalization of TH through advocacy, leadership, and strategic influence. Their engagement ensures that policies are not only technically sound but also aligned with broader health system goals, making regulatory guidelines more practical, sustainable, and responsive to real-world challenges”. Please, this information could be moved to “Discussion” section.
-Lines 294-296: A list of 16 barriers and 16 facilitators have been identified from the included studies described in Error! Reference source not found. and Error! Reference source not found. Please, correct the sentence.
-Lines: 309-313: “This variation in study quality underscores the importance of critically evaluating the evidence base used to inform TH regulatory guidelines. High-quality studies are essential for developing effective, contextually appropriate, and sustainable policies in COPD care.” Please, this information could be moved to “Discussion” section.
-Line 324: “…legalisation.(44-47) Error! Reference source not found.” Please, correct the sentence.
-Table 5 : Which order did the authors use to cite the reports in this table?
Discussion
Line 475: “…As illustrated in Error! Reference source not found., the framework…” Please, correct the sentence.
References
I did not find references from 2025.
Author Response
Reviewer 2
Thank you for the opportunity to review this manuscript. Please find my suggestions and comments.
Subject
Title: “Evaluating the Impact of Regulatory Guidelines on market adoption and Implementation of Telehealth for COPD Patients: A Systematic Literature Review” /Abstract: Conclusion: “Clear, inclusive, and context- sensitive regulatory guidelines are essential to support the successful integration of TH in COPD care” / Lines 399-400: “The findings from the literature demonstrate that there is an effort to establish regulatory guidelines for adopting TH in COPD care in several developed countries”. However, the authors included 15 reports not focused on COPD patients. Please, think about this.
- Thank you for your valuable suggestion. We carefully considered the reviewer’s comment. Following the authors’ discussion, we acknowledge that the 15 reports were not directly linked to COPD care. However, these reports were included because they regulate telehealth systems or evaluate the compliance of telehealth solutions with regulatory standards and policies that encompass COPD care. This reflects one of the knowledge gaps existing in current clinical practice. In the revised manuscript, we have added this clarification to both the Discussion and Limitations sections, along with further insights on the importance of developing disease-specific guidelines for telehealth in COPD.
- The revised manuscript now reads: “However, the literature highlights significant gaps… Existing regulatory guidelines are generic and lack disease-specific guidance.” (Page 2, Lines 78–84).
- The revised manuscript also reads: “Current TH regulations are often general and lack COPD-specific provisions.” (Page 23, Lines 407–410).
- Additionally, the manuscript now states: “Current regulatory guideline reports adopted by health authorities tend to emphasize general operational and technical standards, offering limited guidance for disease-specific applications such as COPD management.” (Page 26, Lines 498–499).
Abstract
Please, describe the date of the study.
- Thank you for your comment. The date of the most recent search strategy has been added to the abstract.
Material and Methods
-Line 96: “Examples of search strategies from all databases…” OR “Search strategies from all databases…” Considering a systematic revision, the authors have to write all search strategies, not only “examples”. Please, think about this.
- Thank you for the suggestion. We have updated the manuscript to include the complete search strategies from all databases. These are now provided in Supplementary Table S1.
-Lines 106-108: “Therefore, the scope of our electronic search was expanded to encompass governmental and non-governmental reports that discuss regulatory guidelines for TH in COPD care or similar contexts. Which similar contexts? I did not find information about this in “reports” included. I found reports about general telehealth. Please think about this.
- Thank you for your comment. This has been corrected in the revised manuscript. (Page 3, Line 114).
-Line 115: STUDIES “…particularly in the context of COPD…” or “…in the context of COPD…” ?
- Thank you for the suggestion. This has been corrected. (Page 5, Line 121)
- I did not find who (the authors) and how the authors identified the studies and reports (They independently evaluated the abstracts....). Considering a systematic revision, the authors have to describe the details about this.
- Thank you for raising this important point. We apologize for this omission. The Methods section has been revised to include details about the study selection process in line with PRISMA 2020 guidelines. The revised manuscript now reads:
“The screening process was conducted in accordance with the PRISMA 2020 guidelines… etc.” (Page 2, Lines 104–109).
Results
-Table 2. Elwyn et al : “…We need more automated…” OR “There is need to …” ? Please, think about this.
- Thank you for your comment. This has been corrected in the revised manuscript.
-Lines 187-192: “The absence of consistent terminology poses challenges for marketing, evaluating, regulating, and implementing TH in COPD care. It also complicates the synthesis of evidence across studies and may hinder the development of unified guidelines and policies. This highlights a pressing need for consensus on terminology and classification in future research and clinical practice to enhance comparability and ensure coherent regulatory and implementation strategies.” Please, this information could be moved to “Discussion” section.
- Thank you. This information has been moved to the Discussion
-Lines 210-217: “Despite the variation in regulatory maturity across healthcare systems, the presence of regulatory guidelines remains a critical enabling for the sustainable and safe deployment of TH in clinical practice. These regulatory guidelines play an essential role in ensuring patient safety, clinical accountability, and interoperability of digital systems. Nevertheless, the heterogeneity of regulatory approaches across healthcare systems highlights the need for greater harmonization through global or regional frameworks. Such alignment would facilitate broader adoption, enhance implementation scalability, and foster international collaboration in TH applications in COPD care.” Please, this information could be moved to “Discussion” section.
- Thank you. This information has been moved to the Discussion
-Lines 246-251: “These individuals or groups serve as catalysts for change, facilitating the institutionalization of TH through advocacy, leadership, and strategic influence. Their engagement ensures that policies are not only technically sound but also aligned with broader health system goals, making regulatory guidelines more practical, sustainable, and responsive to real-world challenges”. Please, this information could be moved to “Discussion” section.
- Thank you. This information has been moved to the Discussion
-Lines 294-296: A list of 16 barriers and 16 facilitators have been identified from the included studies described in Error! Reference source not found. and Error! Reference source not found. Please, correct the sentence.
- Apologies for the inconvenience. This has been corrected in the revised manuscript.
-Lines: 309-313: “This variation in study quality underscores the importance of critically evaluating the evidence base used to inform TH regulatory guidelines. High-quality studies are essential for developing effective, contextually appropriate, and sustainable policies in COPD care.” Please, this information could be moved to “Discussion” section.
- Thank you. This has been moved to the Discussion
-Line 324: “…legalisation.(44-47) Error! Reference source not found.” Please, correct the sentence.
- Apologies for this inconvenience. This has been corrected.
-Table 5 : Which order did the authors use to cite the reports in this table?
- Apologies for this inconvenience. The citation order has been reviewed and corrected in the revised manuscript.
Discussion
Line 475: “…As illustrated in Error! Reference source not found., the framework…” Please, correct the sentence.
- Apologies for this inconvenience. This has been corrected.
References
I did not find references from 2025.
- Thank you for raising this point. The search strategy was updated, and relevant references from 2025 were incorporated where appropriate in the revised manuscript. While no additional eligible primary studies were identified, a 2025 systematic review has been cited in the Discussion to ensure currency and completeness of the literature base.
Round 2
Reviewer 1 Report
Comments and Suggestions for Authors
in better position to publish.
Author Response
We appreciate your feedback. Your insightful comments have significantly improved the clarity, organization, and overall quality of the manuscript.
Reviewer 2 Report
Comments and Suggestions for Authors
Congratulations. Please, find my last comments.
-Lines 107-108: “Initially, all retrieved records were screened by title and abstract to exclude studies that did not meet the inclusion criteria…” OR “Initially, the authors NA and PT independently screened title and abstract to exclude studies that did not meet the inclusion criteria…” Please, think about this.
-Lines 148-149: “…across included studies (n=18)…” Please, “n=18” is a result information. The authors could remove it from "material and methods" section.
-Figure 1: “Identification of studies via other methods”. Which were “other methods”? Please, rewrite this information.
-Table 4 : The authors could write the reference number of each author.
-Table 6 : Which order did the authors use to cite the reports in this table? The authors could organize the information considering the year of publication as presented in Table 3.
Author Response
-Lines 107-108: “Initially, all retrieved records were screened by title and abstract to exclude studies that did not meet the inclusion criteria…” OR “Initially, the authors NA and PT independently screened title and abstract to exclude studies that did not meet the inclusion criteria…” Please, think about this.
- Thank you. This has been corrected as suggested.
-Lines 148-149: “…across included studies (n=18)…” Please, “n=18” is a result information. The authors could remove it from "material and methods" section.
- Thank you. This has been removed from the Methods section.
-Figure 1: “Identification of studies via other methods”. Which were “other methods”? Please, rewrite this information.
- Thank you. This follows the PRISMA guidelines flowchart. We have clarified this in the revised manuscript and also submitted an editable version of the PRISMA flow diagram should any additional modifications be required.
-Table 4 : The authors could write the reference number of each author.
- Thank you. Reference numbers have been added to the table as suggested.
-Table 6 : Which order did the authors use to cite the reports in this table? The authors could organize the information considering the year of publication as presented in Table 3.
- Thank you. We have reordered the information based on the year of publication as suggested.